

**Differentiating local and regional sources of Chinese urban air pollution based on**
**effect of Spring Festival**
Chuan Wang, Xiao-Feng Huang*, Qiao Zhu, Li-Ming Cao, Bin Zhang, Ling-Yan He
Key Laboratory for Urban Habitat Environmental Science and Technology, School of Environment and Energy,
Peking University Shenzhen Graduate School, Shenzhen, 518055, China.
*Corresponding author: huangxf@pku.edu.cn
**Abstract**: The emission of pollutants is extremely reduced during the annual Chinese Spring Festival
(SF) in Shenzhen, China. During the SF, traffic flow drops by ~50% and the industrial plants are almost
entirely shut down in Shenzhen. To characterize the variation in ambient air pollutants due to the
"Spring Festival effect", various gaseous and particulate pollutants were measured in real time in urban
Shenzhen over three consecutive winters (2014–2016). The results indicate that the concentrations of
NOx, volatile organic compounds (VOCs), black carbon (BC), primary organic aerosols, chloride, and
nitrate in submicron aerosols decrease by 50%–80% during the SF period relative to the non-Spring
Festival periods, regardless of meteorological conditions, which suggests that these pollutants are
mostly emitted or secondarily formed from urban local emissions. The concentration decreasing of
regional pollutants or species emitted from natural sources, however, is found to be much less,
especially for bulk $PM_{2.5}$ (–24%) and $O_3$ (6%). More detailed analysis of the Spring Festival effect
reveals an urgent need to reduce emissions of $SO_2$ and VOCs on a regional scale rather than on an
urban scale to reduce urban $PM_{2.5}$ in Shenzhen, which can also produce some use for reference for
other megacities in China.




**Key words**: Spring Festival effect; local emissions; regional pollution; PM₂.₅; ozone

**1 Introduction**

The rapid economic development and urbanization of China over the recent decades has brought with it the consequence of severe atmospheric pollution, especially in the key economically developed regions, such as the Beijing–Tianjin–Hebei region (Sun et al., 2013, 2015; Guo et al., 2014), the Yangtze River Delta (Huang et al., 2013), and the Pearl River Delta (PRD), as well as their densely populated megacities (Hagler et al., 2006; Zhang et al., 2008; He et al., 2011). Great efforts have been made to determine the sources and formation mechanisms of fine particles (PM$_{2.5}$) in these region. Previous studies indicate that PM$_{2.5}$ forms from primary fine particles and through secondary formation from gaseous precursors (Zhang et al., 2008; Zheng et al., 2009; Huang et al., 2014), and the sources of local production and regional transport are both important (Huang et al., 2014; Huang et al., 2006, 2011; Li et al., 2015).

The causes of air pollution in urban atmosphere in China are particularly complicated, and bring great challenges to management strategies for protecting human health (Parrish and Zhu, 2009). To explore the causes of urban air pollution in China, previous studies have focused on monitoring and comparing the reduction in emissions during special events, such as the 2008 Beijing Olympic Games (Huang et al., 2010), the 2010 Guangzhou Asian Games (Xu et al., 2013), the 2014 Asia Pacific Economic Cooperation conference (APEC) (Chen et al., 2015; Sun et al., 2016; Zhang et al., 2016) and the 2015 China victory day parade (Zhao et al., 2016). During such events, the air quality improved remarkable because of short-term limitations on traffic and industrial activity (Huang et al., 2010; Wang et al., 2010; Xu et al., 2013; Sun et al., 2016; Zhao et al., 2016). However, these limitations were temporary,





non-repeatable measures, so the reported emission reductions cannot be verified. Actually, a
spontaneous reduction in emissions occurs every year in China during the Spring Festival (SF), which
is the single most important holiday in China. During the week-long holiday (in January or February
every year), the urban emission patterns depart significantly from the usual patterns: traffic decreases
in the mega cities because most people are not working, and most of the industries, stores, and
production sites are closed in the city except for the utilities and industries (e.g., power plants) that
cannot be shut down (Qin et al., 2004; Feng et al., 2012; Shi et al., 2014). Tan et al. (2009) reported
that the concentrations of NOx, CO, NMHC, $SO_2$, and $PM_{10}$ were lower in the SF periods than in the
non-Spring Festival (NSF) periods in the metropolitan area of Taipei over 1994-2006, while the
variation of $O_3$ was in a reversed trend. Jiang et al. (2015) found that the ambient concentrations of
VOCs had a sharp decline by ~60% during the SF in Shijiazhuang.

This study focuses on Shenzhen as a special example to evaluate the effect on urban air pollution of
the SF. Shenzhen is in the eastern Pearl River Delta (PRD) and is the fourth largest economic center
in China, with a total residential population of over 10 million and a fleet of civilian vehicles of more
than 3.1 million (Shenzhen Yearbook of Statistics, 2015). Known as the country's city of most floating
population, Shenzhen owns 7.4 million immigrants in 2014, which accounts 70% of the city's total
population (Shenzhen Yearbook of Statistics, 2015). During the SF period, over 50% of the residents
in Shenzhen are used to travel back to their hometowns (http://sz.gov.cn). It is reported that the traffic
flow in Shenzhen during the SF of 2016 (Feb 7–13) was only the half before the SF period
(http://sz.gov.cn). Additionally, industrial activities are almost totally suspended in Shenzhen during
the SF period. To characterize the air quality during such extreme reductions of anthropogenic



activities during the SF period in Shenzhen, various air pollutants in Shenzhen urban areas were
comprehensively and systematically monitored in real time in winter for three consecutive years
(2014–2016). The annual SF in Shenzhen thus provides an excellent spontaneous control experiment
for local emissions, which could provide unique and valuable information regarding the sources of
urban air pollution.

**2 Experimental methods**
**2.1 Monitoring sites and meteorological conditions**
The monitoring site (22°36′N, 113°54′E) was on the roof (20 m above ground level) of an academic
building on the campus of Peking University Shenzhen Graduate School (PKUSZ) (Figure S1). No
significant anthropogenic emission sources exist nearby. The sampling schedule ran roughly from late
January to early March over 2014–2016, which includes the official SF holiday period and the prior
and following periods. Our definition of the SF period follows that of the statutory public holiday
calendar in China, and it is continuous seven days in each year. While the seven days immediately
before or after the holidays are actually the transition periods between the holidays and normal days
(called the Tran. periods hereafter), when people begin to move from the city (or their hometowns) to
their hometowns (or the city), the typical non-spring festival (NSF) periods are better defined as the
7–14 days close to the SF period (called the NSFT period hereafter, where T indicates time similar).
The specific dates and the average meteorological parameters are listed in Table 1, and Figure S2
shows wind rose plots. The data in Table 1 show that the meteorology differs among the SF, NSFT,
and Tran. periods. To control for the influence of meteorology on the evaluation of emissions, we
selected another 7-day period each year when the meteorology is similar to that of the SF period (called



the NSFM period hereafter, where M indicates meteorology similar); the detailed parameters are listed
in Table 1 and Figure S2. The meteorological data for the SF period are fairly similar to those of the
NSFM period, suggesting similar meteorological conditions.

**Table 1.** Summary of meteorological conditions at sampling site during the SF, NSFT, NSFM and
Tran. periods of 2014–2016.

| | | SF | Tran. | NSFT | NSFM |
|---|---|---|---|---|---|
| Data period | 2014 | Jan 31–Feb 6 | Feb 7–Feb 13 | Feb 14–Feb 20 | Feb 20–Feb 26 |
| | 2015 | Feb 18–Feb 24 | Feb 11–Feb17 | Feb 4–Feb10 | Jan 24–Jan 30 |
| | 2016 | Feb 7–Feb 13 | Feb 14–Feb 20 | Feb 21–Feb 27 | Feb 27–Mar 4 |
| Meteorological parameters | Temperature (℃) | 19.0±4.7 | 14.1±5.3 | 14.1±4.0 | 18.1±3.8 |
| | RH (%) | 68.1 ±17.8 | 69.3±18.4 | 64.9±16.7 | 67.4±14.7 |
| | Wind speed (m s$^{-1}$) | 0.88 ±0.57 | 0.81±0.49 | 0.83±0.48 | 0.86±0.55 |
| | Dominant wind direction | NW | NW and NE | NW and NE | NW |
| | Precipitation (mm) | 0 | 0 | 0 | 0 |
| | UVA (W m$^{-2}$) | 5.4±8.5 | 2.5±4.3 | 3.8±6.7 | 5.0±8.0 |
| | UVB (W m$^{-2}$) | 0.24±0.40 | 0.11±0.25 | 0.16±0.32 | 0.22±0.38 |



## 2.2 Instrumentation


For the ambient sampling in this study, the measuring instruments were placed in a room on the top
floor of a four-story teaching building at PKUSZ. A high-sensitivity proton transfer reaction mass
spectrometer (PTR–MS) (Ionicon Analytik GmbH, Austria) was used to measure the selected volatile
organic compounds (VOCs). The PTR–MS measured a total of 25 masses in the selected ion mode at
a time resolution of 30 s. Background checks were done for 30 of every 300 scan cycles with an
activated charcoal trap at 360 °C, which can remove VOCs from the ambient air without changing
water content. The VOCs reported here (Table S1) may be broadly classified into three categories:
oxygenated VOCs [OVOCs: methanol, acetone, methyl ethyl ketone (MEK), acetaldehyde, and acetic
acid], aromatics (benzene, toluene, styrene, C8 and C9 aromatics), and three types of tracers [isoprene,
acetonitrile, and dimethyl sulfide (DMS)]. The PTR–MS was calibrated every 5 to 7 days by using a
TO15 mixture standard (Air Environmental Inc., US) and permeation tubes (Valco Instruments Co.
Inc., US) (de Gouw and Warneke, 2007).

An aerodyne high-resolution time-of-flight aerosol mass spectrometer (HR–ToF–AMS) (Aerodyne
Research, US) was deployed to measure non-refractory $PM_1$ (NR-$PM_1$) (Canagaratna et al., 2007) in
the period 2014–2015 with a time resolution of 4 min. An aerosol chemical speciation monitor (ACSM)
(Aerodyne Research, US) was used in 2016 with a dynamic resolution of 10 min. The detailed
description of the ACSM is available in the recent review (Ng et al., 2011). The HR-ToF-AMS and
ACSM were calibrated every month following the standard protocols (Ng et al., 2011; Jayne et al.,

117  2000).






An aethalometer (AE–31) (Magee, US) was used for simultaneous detection of refractory black carbon
(BC) with a time resolution of 5 min. In addition, a Scan Mobility Particle Sizer (TSI Inc., US) system
was used to determine the particle number size distribution in the size range 15–615 nm (Stokes
diameter) with a time resolution of 5 min. The stokes diameters of 15–615 nm is converted to
aerodynamic diameters of 22–800 nm, and then $PM_{0.8}$ mass concentration can be calculated with the
particle density assumed according to the AMS measurement results of species.

To measure the $PM_{2.5}$ mass concentration, we used a Thermo Scientific TEOM 1405–D monitor. The
trace-gas instruments included a 43i sulfur dioxide ($SO_2$) analyzer, a 42i nitric oxide (NO)–nitrogen
dioxide ($NO_2$)–nitrogen oxide (NOx) analyzer, a 49i ozone ($O_3$) analyzer, and a 48i carbon monoxide
(CO) analyzer (Thermo Scientific, US). A meteorological station, also located on the roof of the same
building, measured the main meteorological parameters, such as temperature, relative humidity, and
wind speed (see Table 1).

**3 Results and Discussion**
**3.1 The NSF–SF differences for major air pollutants**
The results of observations from 2014 to 2016 appear in Figures S3–S5. Figure 1 shows the averaged
percent changes in the concentrations of major air pollutants of the SF periods relative to the two NSF
periods and Tran. period over 2014–2016. The compounds m/z 44 and m/z 57 are the tracer of
oxygenated organic aerosol and the tracer of primary hydrocarbon organic aerosol (Zhang et al., 2005),
respectively, which are organic fragments in the AMS measurements. The notation $O_3$–8h refers to the
average maximum $O_3$ concentration over a continuous diurnal 8 h and $PM_{0.8–2.5}$ refers to the difference



between the concentrations of $PM_{2.5}$ and $PM_{0.8}$.

We can divide these air pollutants into three classes based on their percent changes: The group with
the largest drop (hereinafter called "LD") in concentration includes the aromatics (−50% to −88% for
the various species, see Figure S6), OVOCs (−40% to −85% for the various species, see Figure S6),
NOx, chloride (Chl), nitrate ($NO_3^-$), BC, and m/z 57. The concentrations of these pollutants all
decrease by over 50% during the SF period compared with both the NSF periods. Apparently, the
dominant sources for most of these pollutants are primary local emissions, such as combustion sources
for BC, m/z 57, and NOx (Zhang et al., 2005; Kuhlbusch et al., 1998; Lan et al., 2011), and vehicle,
industrial and solvent use for aromatics (Liu et al., 2008). As detailed in the following section, the
diurnal patterns and relationships with respect to wind speed further confirm the sources of these
pollutants. The dramatic decrease in the ambient concentrations of these species is consistent with
reduction in local anthropogenic activities in Shenzhen during the SF period. The SF causes a 50%
decrease in urban traffic and temporarily closing of almost all local industrial plants. The nitrate and
chloride measured by AMS or ACSM are actually ammonium nitrate ($NH_4NO_3$) and ammonium
chloride ($NH_4Cl$), which are typical secondary air pollutants. These are thought to form via reversible
phase equilibria with gaseous ammonia ($NH_3$), nitric acid ($HNO_3$), and hydrochloric acid (HCl) (He et
al., 2011; Huang et al., 2011; Zhang et al., 2007). Typically, the formation of $NH_4NO_3$ from NOx and
the reaction between HCl and $NH_3$ occur quickly in the atmosphere (Stelson and Seinfeld, 1982; Baek
et al., 2004), suggesting that the concentrations of $NO_3^-$ and Chl in winter in Shenzhen depend largely
on the emission of precursors such as HCl and NOx. Therefore, the significant decline in the ambient
concentrations of $NO_3^-$ and Chl during the SF period and indicates that their precursors also have local



origins, similar to the case for primary pollutants (this is also supported by the discussion in the
following sections). The huge decline in the ambient concentration of OVOCs during the SF period
shows that the source of these pollutants is (i) mainly from local emissions, including vehicle and
industrial emissions (Schauer et al., 1999; Singh et al., 2001) and (ii) from secondary reactions
involving local primary VOCs (Liu et al., 2015). Thus, in the LD group, the significant reduction in
local sources of pollutants strongly impacts the concentration of air pollutants.

The pollutants in the next group undergo a medium drop in concentration during the SF period
(hereinafter called "MD"). These are $PM_{2.5}$, NR–$PM_1$, $PM_{0.8}$, organic aerosol, m/z 44, sulfate ($SO_4^{2-}$),
ammonium ($NH_4^+$), isoprene, acetonitrile, DMS, and carbon monoxide (CO), and their percent change
varies from −20% to −55% when comparing the SF periods to the NSFT and NSFM periods. The
species in this group are either typical secondary regional air pollutants, such as CO, which has a long
lifetime and is a tracer for combustion sources, acetonitrile from rural biomass burning (de Gouw et
al., 2003; Le Breton et al., 2013), m/z 44 representing secondary organic aerosols, $SO_4^{2-}$ from $SO_2$
oxidation (He et al., 2011; Huang et al., 2011), or typical tracers for natural sources, such as isoprene
from vegetation (Guenther et al., 1995) and DMS from marine source (Dacey and Wakeham, 1986).
In winter, the northeastern monsoon prevails in the PRD and transports significant amounts of various
air pollutants from the northern inland, increasing air pollution of the PRD to the highest levels through
the year (Huang et al., 2014). In particular, the small drop in CO concentration during the SF period
puts it in this group and indicates that the contribution to regional air pollution does not decrease
significantly during the SF period. Note that, the significant declines of the concentrations of isoprene
and DMS imply that they have anthropogenic sources, which will be supported in the following



sections. The other air pollutants in this group are the reflection of the overall effect of the reduction
of relevant air pollutants: OA is the whole of the two types of organic aerosol represented by m/z 44
and m/z 57, $NH_4^+$ is represented by $SO_4^{2-}$, $NO_3^-$ and Chl, and NR–$PM_1$ is the sum of all species
measured by AMS or ACSM (their average chemical compositions during different periods are shown
in Figure S3–S5).

The group of pollutants with smallest decrease in concentration (hereinafter called "SD") includes $SO_2$,
$PM_{0.8–2.5}$, and $O_3$. The magnitude of the average percent change is less than 20% relative to the two
NSF periods. The average concentration of $SO_2$ was only 2.8 ppbv in Shenzhen in 2015
(http://www.szhec.gov.cn/), which is much lower than that in Beijing (4.7 ppbv) and elsewhere in
China (http://www.zhb.gov.cn/). This result is partly attributed to the negligible coal consumption in
Shenzhen, which instead relies mainly on natural gas and liquefied petroleum gas (Shenzhen Yearbook
of Statistics, 2015). The emission inventory indicates that international marine container vessels are
the dominant source of $SO_2$ in Shenzhen (Wang et al., 2009), and according to official statistics, the
Shenzhen port piloted 401, 568, and 521 ships during the SF period in the years 2014–2016,
respectively, which is quite similar to numbers for the NSF periods (http://www.pilot.com.cn).
Obviously, the small decrease of $SO_2$ is reasonably related with the stable ship emissions during the
SF periods. The small decline of $PM_{0.8–2.5}$ during the SF period suggests that the reduction of more
aged particles of lager sizes in $PM_{2.5}$ is much lower than fresher particles of smaller size. This can be
also confirmed by particle number concentration (PNC) measurement by SMPS, as shown in Figure
2. The largest difference of the PNC between the SF and NSFM periods exists mainly in a smaller size
range (20–40 nm), which is recognized as the nucleation mode or second Aitken mode that represents





fresh combustion emission (Ferin et al., 1990). Contrary to other pollutants, the concentrations of $O_3$,
present small increasing during the SF period (except a little decline when comparing $O_3$–8h with the
NSFM period), which could be attributed to the different drop rates for $O_3$ precursor species, i.e. NOx
and VOCs (Qin et al., 2004), and will be discussed in more detail in section 3.2.

The results for the various species during the NSFT and NSFM periods are consistent, which suggest
that meteorology has only a small impact on their concentrations. This means that the strong decrease
in the concentrations of air pollutants in group LD and MD is mainly due to the abatement of local
sources. The larger decline in the SF period when compared to NSFT than to NSFM is associated with
the lower temperature and stronger winds from the polluted northwest inland of the PRD during the
NSFT period. In addition, the effect of the SF on the concentrations of the various species is almost
identical each year (see Figure S7), which further confirms that the pollutant concentrations are
determined primarily by the activity of the sources. In Figure 1, the percent changes of pollutants of
the SF periods relative to the Tran. periods are also presented, and it is found that the three-group
classification defined above is also applicable, while the decrease levels are lower. For example, the
average decrease percent of Group LD for the Tran. period case is 61%, while those for the NSFT and
NSFM cases are 71% and 63%, respectively. This result is consistent with the fact that the SF travel
of people occurred mostly during the seven days before and after the SF holidays (http://sz.gov.cn),
and thus the city became much emptier even in the Tran. periods. In order to make a deeper and valid
comparison for revealing the SF effect, the following discussion will only take the NSFM periods and
SF periods for comparative analysis due to their more similar meteorology.




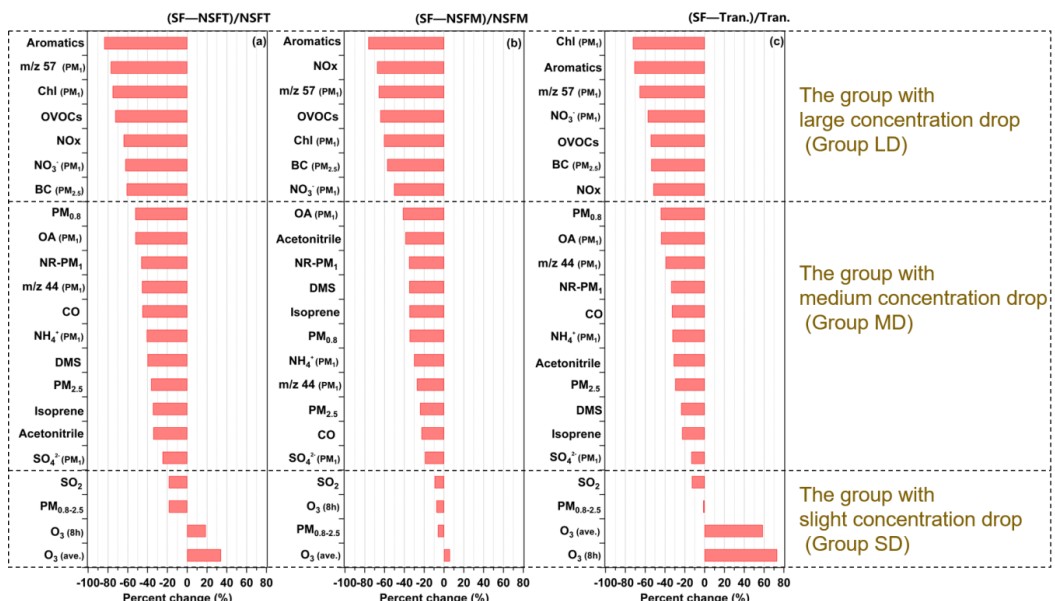


**Figure 1.** Percent change in concentrations of major air pollutants during the SF period relative to (a)

Tran., (b) NSFT and (c) NSFM periods averaged over 2014–2016.

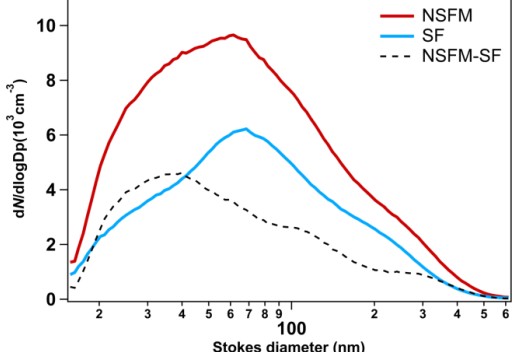


**Figure 2.** Distribution of particle number concentration in the 15–615 nm size range during the SF

and NSFM periods.






### 3.2 The diurnal variation of major air pollutants


As shown in Figure 3, the diurnal cycles of all LD pollutants (except for the OVOCs) reveal significant
peaks in concentration around 8–9 am in the NSFM period, which is attributed to the low planetary
boundary layer (PBL) in the morning and local rush hour traffic emissions. The evening rush hour
peak, however, is not apparent for all the species, which is attributed to the higher ambient temperature
and thus the higher PBL at that time than in the morning. During the SF period, the concentrations of
all pollutants are far lower over the entire day. In particular, the rush-hour peaks become much smaller
or disappear altogether, which is consistent with the large reduction in local vehicle emissions during
the SF period. Although the sources of Chl remained uncertain in previous studies (Huang et al., 2011;
Aiken et al., 2008), the maximal reduction (80%) in this pollutant during the morning rush hour during
the SF period implies that local traffic emissions account for a significantly fraction of this pollutant
in Shenzhen (Figure 3E). Contrary to other species in this group, the concentration of OVOCs is high
in the daytime and peaks in the morning after the morning rush hour time during the NSFM period
(Figure 3D), suggesting that photochemical production and/or daytime industrial activities may be
important sources of OVOCs. The concentrations of different aromatics and OVOCs usually follow
similar diurnal variations (Figure S8).

The diurnal variations of the MD pollutants are relatively smooth except for the two natural VOCs
(isoprene and DMS; see Figures 3L and 3M), which indicates that these pollutants come from regional
sources and are dispersed more uniformly over a larger scale. The apparent difference of the diurnal
variations of those anthropogenic air pollutants between the SF and NSFM periods also exists in the
rush hours (except for acetonitrile; Figure 3J), however, the reduction in local sources has a relatively





weak effect on the overall concentrations of these pollutants. Acetonitrile, which is a tracer of biomass
burning, is more concentrated during the daytime and its peak concentration occurs after the rush hours
during the NSFM period (Figure 3J), which is similar to the result obtained for OVOCs and may be
attributed to the influence of daytime anthropogenic activities, for example, industrial biomass boilers.
Isoprene is primarily emitted by vegetation as a function of light and temperature, so the concentration
of this pollutant goes through a broad peak that spans the daytime hours during both the NSFM and
SF periods. The percent change in isoprene concentration between the SF and NSFM periods is
approximately −40% (Figure 3M), despite the NSFM and SF periods having similar temperature and
solar radiation, which implies that the contribution of anthropogenic sources to isoprene cannot be
overlooked in Shenzhen. Many studies have reported isoprene from vehicle exhaust, especially in cold
seasons (Barletta et al., 2005; Borbon et al., 2001). DMS is reported to be a marine tracer (Dacey et
al., 1986), its peak concentration occurs in the morning during both the NSFM and SF periods (Figure
3L), which is presumably related to the minimal PBL. The concentration of DMS decreases by 30%–
50% during the SF period, which reflects the reduced DMS emissions from anthropogenic sources. As
reported in the literature, industrial activities can make significant emissions of DMS (Schafer et al.,

272    2010).


The diurnal variations of $PM_{0.8-2.5}$, $SO_2$ and $O_3$ demonstrated more similar concentrations and trends
in the SF and NSFM periods, respectively (Figure 3T–3V). For $PM_{0.8-2.5}$, a small difference is found
in the afternoon, which is supposed to be a result of more aged larger particles formed through stronger
photochemical reactions during the NSFM period. Though, slight differences appear in $SO_2$
concentration, mainly during the nighttime when the PBL is low. These data suggest a minor role of





local near-ground $SO_2$ sources, such as vehicles. Although the daytime peak concentration of $O_3$ during
the NSFM period is slightly greater than that during the SF period, this trend reverses from the evening
to the midmorning hours. Similar phenomena have also been observed in other emission-reduction
studies of urban areas (i.e., emissions are greater on holidays than on non-holidays) (Qin et al., 2004;
Tan et al., 2009). In addition, emissions were higher during the 2008 Beijing Olympic Games (Chou
et al., 2011), during which strict controls were imposed. A recent study reported that, in most of the
PRD region, $O_3$ formation is VOC limited in the morning and becomes NOx limited during peak $O_3$
hours (Li et al., 2013). The concentrations of NOx and VOCs decrease gradually from 8 to 12 h during
the SF period whereas the concentration ratio TVOC/NOx increases (see Figure 3H, where TVOC
concentration is the sum of aromatic and OVOC concentrations). Thus, the lack of NOx at noon during
the SF period hinders the generation of $O_3$. At other hours in the NSFM period, a higher NO
concentration destroys $O_3$, implying that the oxidation reaction with NO may produce a titration effect
(Qin et al., 2004; Tan et al., 2009). As a result, although the reduction in emissions of urban
anthropogenic sources leads to a significant decline of NOx and VOCs, this reduction does not mitigate
the average ambient $O_3$ concentration, which implies that the concentration ratio VOC/NOx plays an
important role in controlling $O_3$ concentration.



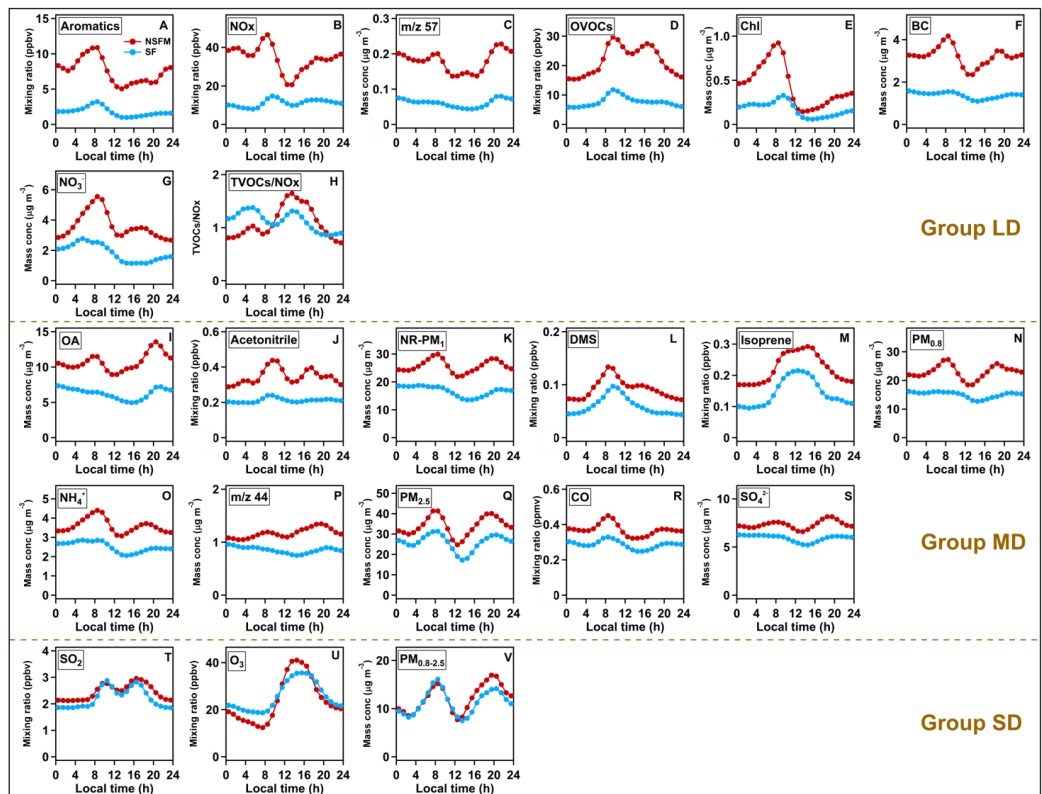

**Figure 3.** Diurnal variations in concentrations of major air pollutants at PKUSZ site over the SF

(blue dots) and NSFM (red dots) periods.

## 3.3 Influence of wind on observed air pollutants

Wind plays a crucial role in the dilution and transport of air pollution. The wind field is essentially the

same during the SF and NSFM periods. In general, the concentrations of LD air pollutants depend

strongly on wind speed during the NSFM period, whereas this dependence becomes much weaker

during the SF period (Figure 4). The difference in the concentration of LD air pollutants (including

various aromatics and OVOCs, see Figure S9) between the NSFM and SF periods is maximal (50%–

80%) under conditions of low wind speeds (<1 m/s) because local pollution can more easily

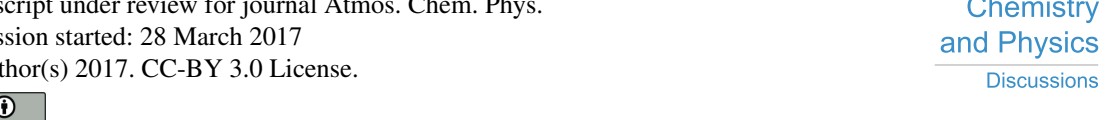

accumulate under these conditions. These results confirm that the concentration of air pollutants from
local sources is strongly reduced during in the SF period.

Compared with the LD pollutants, the concentrations of CO, $SO_4^{2-}$, m/z 44, isoprene, DMS, and
acetonitrile do not vary significantly with wind speed during the NSFM period, providing further
evidence that these pollutants come from regional or natural sources and are consequently more evenly
distributed in the atmosphere.

In the Group SD, $SO_2$ is generally little influenced by wind speed during the SF period, while some
higher concentrations appeared under low wind speeds during the NSFM period, indicating again small
contribution of urban local sources to $SO_2$. The fluctuation of $PM_{0.8–2.5}$ both in the SF and NSFM
periods does not reveal a clear relationship with wind speed, suggesting again it is not a typical locally
emitted air pollutant. The variations of $O_3$–8h display the opposite trend to other air pollutants both in
the SF and NSFM periods, growing smoothly as wind speed increases, which could be possibly
attributed to more regional transport and/or the higher VOCs/NOx ratio under high wind speeds
(Figure 4H). Note that, when the proportion of regional transport relative to local emission becomes
bigger under higher wind speeds, the concentrations of $NO_3^-$, $SO_4^{2-}$, m/z 44, $PM_{0.8–2.5}$, and $O_3$–8h are
even slightly higher in the SF period than in the NSFM period, implying that regional photochemical
production during the SF period is not weakened.





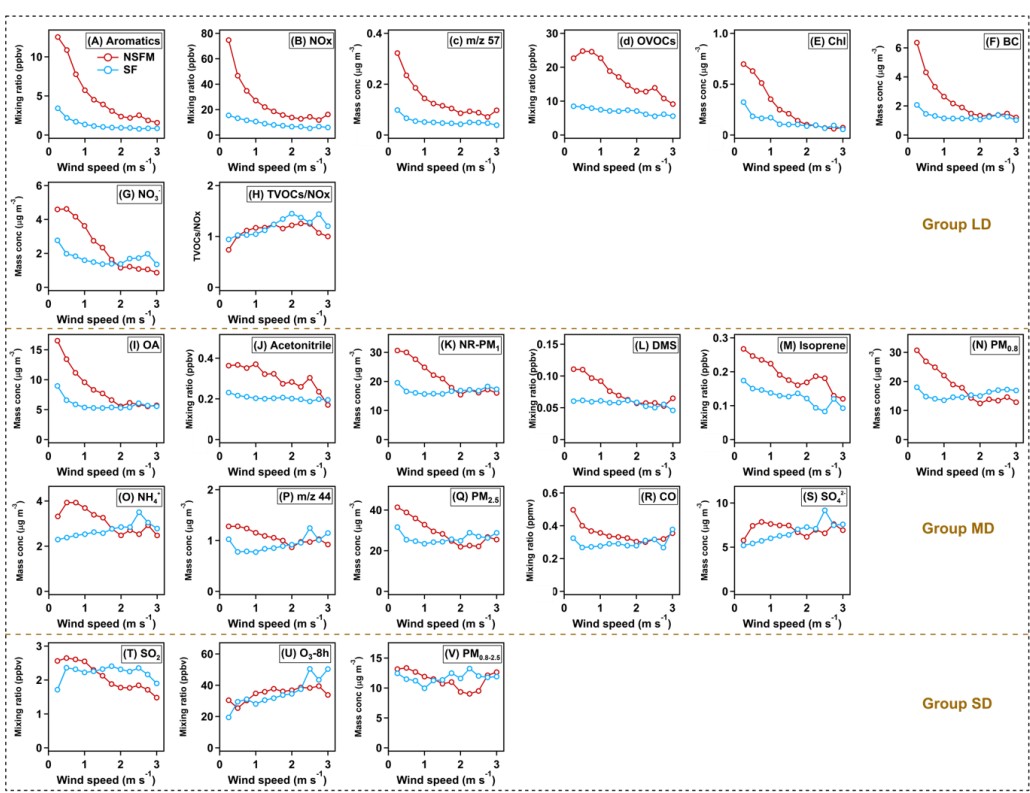

**Figure 4.** Concentrations of major air pollutants as a function of wind speed during the SF and

NSFM periods.

### 3.4 Emission ratio analysis

Several groups of special correlations were applied to analyze the source characteristics of air

pollutants in Figure 5. CO and BC are both products of incomplete combustion (Subramanian et al.,

2010), but gaseous CO can travel farther because of its longer atmospheric lifetime (approximately a

month for CO vs a week for BC) (Khalil et al., 1990; Ogren et al., 1983). As shown in Figure 5A, the

correlation coefficient and slope between BC and CO during the NSFM period ($r^2 = 0.56$, slope = 13.9)



is greater than during the SF period ($r^2 = 0.31$, slope = 3.9), suggesting that local combustion sources
make a much greater contribution during the NSFM period, but decline significantly during the SF
period (He et al., 2011). The concentrations of two natural species, isoprene and DMS, are not
correlated with CO during the SF, whereas their correlation with CO is non-negligible during the
NSFM period (Figures 5B and 5C), suggesting again that these pollutants have an anthropogenic
source during the NSFM period.

The toluene/benzene ratio can be used to estimate the contribution of traffic emissions (Schneider et
al., 2005). Generally, a value of 1.2–3 is found to be characteristic of vehicular emission in many urban
areas (Nelson et al., 1984; Wang et al., 2002; Araizaga et al., 2013). The lower ratio of toluene to
benzene (ave.=2.6) in the SF period suggests that the dominant source is vehicle emission. This ratio
in the NSFM period, however, is much higher (ave.=7.7), indicating more complicated sources of
VOCs like huge amount of toluene solvent usage in industrial activities in PRD (Barletta et al., 2005,
2008; Chan et al., 2006). This finding is well consistent with the temporary closure of industrial plants
in the SF period, which leads to little toluene emission.



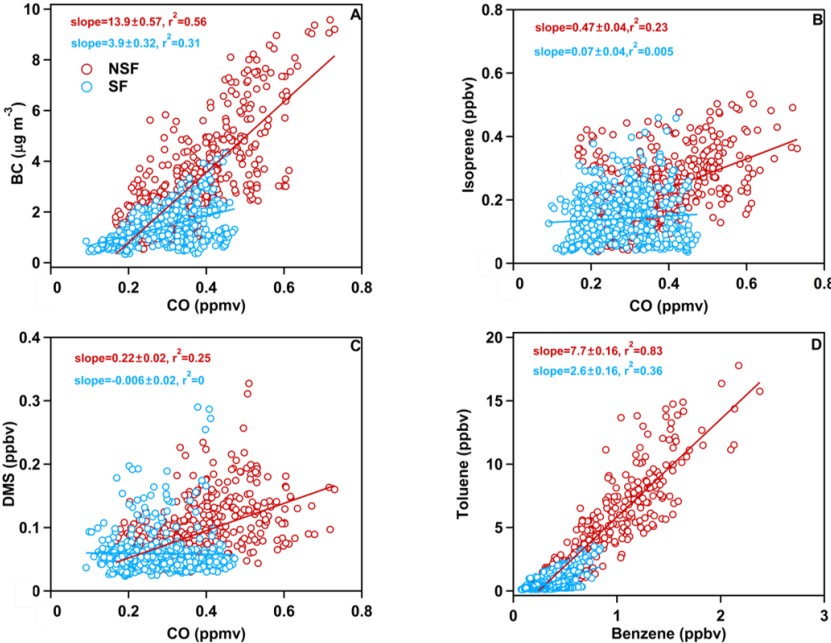

**Figure 5.** Correlation between air pollutants (A) BC and CO (B) isoprene and CO, (C) DMS and CO, and (D) toluene and benzene during the SF (blue circles) and the NSFM (red circles) periods.

**3.5 Conclusions**

This study uses the SF in Shenzhen to investigate how the urban air quality reacts to significant, temporary reductions in emission. During the winters of 2014 to 2016, the air quality was observed continuously at Peking University Shenzhen Graduate School, from which we obtained the percent change in the concentrations of various air pollutants during the SF periods with respect to the comparable NSF periods. The analysis of these data shows that, despite meteorological variations, the Spring Festival clearly and consistently influences the urban concentrations of various air pollutants. The air pollutants can be divided into three groups: the large-decrease (LD) pollutants are those with





a percent change in concentration of −50% to −80% during the SF period and include aromatics, NOx,
m/z 57, OVOCs, Chl, BC, and $NO_3^-$. These results are consistent with the variation in urban emission
sources during the SF, suggesting that these pollutants are mostly directly emitted or formed from
secondary reactions between locally emitted pollutants. The medium-decrease (MD) pollutants are
$PM_{2.5}$, NR–$PM_1$, $PM_{0.8}$, organic aerosol, m/z 44, $SO_4^{2-}$, $NH_4^+$, isoprene, acetonitrile, DMS, and CO;
the concentrations of these pollutants decrease by 20% to 55% during the SF, which indicates that the
extreme reduction in urban emissions during the SF period has limited effect on regional or natural air
pollutants. These results provide further evidence that the origins of these pollutants are primarily
regional or natural. Finally, the slight-decrease (SD) pollutants include $SO_2$, $PM_{0.8-2.5}$, and $O_3$. The
average percent change in the concentrations of these pollutants during the SF period is less than 20%,
which indicates that a significant reduction in urban emissions does not significantly affect their
concentration. Of particular interest is the origin of $PM_{0.8-2.5}$, which is almost completely regional. In
addition, it is found that the concentration of $O_3$–8h correlates strongly with the concentration ratio
TVOC/NOx.

The results of this study show that the extreme reductions in urban emissions of Shenzhen only affects
the concentration of smaller fresh particles, such as $PM_{0.8}$, whereas the reduction of $PM_{2.5}$ is only
slightly affected because of the weak influence on aged, larger particles such as $PM_{0.8-2.5}$. The
concentrations of $SO_4^{2-}$ and secondary organic aerosols are hardly unaffected by local reductions in
emissions. Therefore, reducing the emissions of $SO_2$ and VOCs on a regional scale is critical for
reducing their concentrations and achieving the goal of reducing concentrations of $PM_{2.5}$, at least for
South China. On the other hand, $O_3$ has recently become an increasingly important air pollutant in



China, especially in the PRD. However, the significant reduction of the concentration of its precursors
(NOx and VOCs) during the SF period does not lead to significant reduction of the $O_3$ concentration
because of the concentration ratio VOCs/NOx remains unchanged. Consequently, further
investigations are required to control not only the emissions of VOCs and NOx but also their
concentration ratio.

**Acknowledgements**
This work was supported by the National Natural Science Foundation of China (U1301234 &
41622304), the Ministry of Science and Technology of China (2014BAC21B03), and the Science and
Technology Plan of Shenzhen Municipality.

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
