# Peer review of "Differentiating local and regional sources of Chinese urban air pollution based on effect of Spring Festival"

_Atmospheric Chemistry and Physics, 2017_

## Referee Comment (RC1) · Anonymous Referee #2 · 19 Apr 2017

General comments:

Large reduction in emissions of anthropogenic pollutants in a mega city during the week-long holiday gives an opportunity to investigate how the urban air quality reacts to reductions in emission. This study conducted a systematical examination of the "Spring Festival effect" over three consecutive winters in Shenzhen with a population of greater than ten million, and the difference in the concentrations of various air pollutants between the Spring Festival (SF) and non-Spring Festival (NSF) periods was exploited to indicate that the origins of pollutants are primarily local or regional.

Although "holiday effect" has been discussed in a large number of studies in recent decades, the rich and comprehensive dataset provided in this study is informative and

helpful for understanding of sources of a large array of species with considerable differences in properties. The authors clearly show their own contribution in the study of Spring Festival effect. In the present form, the authors focus on exploiting the percent change in the concentrations of various air pollutants to differentiate contribution from local and regional sources. The value of this paper could be further enhanced if the authors can make more in-depth discussion on the species (e.g., PM0.8–2.5 and O3), which had a small difference between the SF and NSF periods. It is interesting to know that these species revealed only a small difference when traffic flow dropped by ∼50% and the industrial plants were almost entirely shut down. It is well known that PM has numerous and complex constituents contributed from diversified sources (primary and secondary, anthropogenic and natural), and O3 is a secondary photochemical product with nonlinear relation with its precursors NOx and VOCs (anthropogenic and natural). I encourage the author to strengthen the link between their observation results and the possible causes (composition, property, sources, transport, physical and chemical processing, meteorology, etc.). In general, the manuscript was well written and organized. The subject of the paper is well within the scope of ACP. The paper is suitable for publication provided the general comments and following points are addressed.

Specific comments:

Page 7 Lines 137-139: 2016. Are "all fragments" of m/z 44 and m/z 57 measured by AMS are the tracers of oxygenated organic aerosol and primary hydrocarbon organic aerosol, respectively? Or "most of fragments" are?

Page 11 Lines 212-213: This sentence is confusing to me. In addition to emissions and sinks, the concentration of air pollutants is also dominated by meteorological conditions, especially PBL and wind field. In Table 1, meteorological conditions in NSFT and NSFM are not much different (similar wind speed and no precipitation), and it may be the major reason for the small difference in the concentrations of most air pollutants between the two periods. It is not suitable to conclude that meteorology has only a small impact on their concentrations. Please provide a more appropriate interpreta-
tion.

Page 15 Lines 282-283: "emissions are greater on holidays than on non-holidays" and "In addition, emissions were higher during the 2008 Beijing Olympic Games" Do the authors mean "emissions of O3"? If so, I suggest using the word "concentrations" instead of "emissions".

Page 15 Lines 287 and 288: VOC/NOx ratio gives an important idea that it tends towards a NOx-sensitive or VOC-sensitive environment. However, VOCs are a very complex mixture of compounds with large difference in reactivity with respect to ozone formation. The true impact of VOCs to ozone formation is more relevant to the total reactivities of VOC species rather than to the total amount of VOCs. Furthermore, the VOCs reported in the study (Table S1, measured by PTR–MS) include only a total of 13 masses. TVOC/NOx ratio in the study should be carefully used to explain its effect on the ozone level due to above-mentioned concerns.

Page 15 Lines 288 and 289: Do the authors mean the chemical regime at noon during the SF period was NOx-sensitive? If so, the authors should provide a clearer explanation to support the statement.

---

## Referee Comment (RC2) · Anonymous Referee #3 · 29 May 2017

The paper presents an assessment of the effect of the Chinese Spring Festival on urban air quality in a southern China city. Various trace gases and aerosols were measured over three consecutive winters (2014-2016), including both the Spring Festival (SF) and non-Spring Festival (non-SF) periods, at an urban site of Shenzhen city. By comparing the concentrations of these pollutants in different periods, the authors show that the decreases for some pollutants are significant (by 50%-80%) in the SF with respect to the non-SF while the decreases of others pollutants are smaller. They suggest that such differences in the reduction extent of pollutant concentrations be used to judge their sources, i.e., predominantly from local emissions or from regional transport. This study, together with a valuable dataset, should be a welcome addition to the

literatures on the holiday effects on air quality. The manuscript can be accepted for publication after the following issues have been well addressed.

As the main purpose of the study is to distinguish the sources of urban pollution between the local emissions and regional transport, the local and regional pollution sources should have been clearly defined. It would be great if the authors could provide a map showing the geographical distributions of emissions rates of major air pollutants in Shenzhen city and its surrounding areas during wintertime. It is also suggested that if possible, the geographical extent of pollution reduction during the SF in relative to the non-SF period be identified, perhaps by using population density or satellite product.

The study identifies the primary source of each pollutant, i.e. the local or regional origin, using the reduction extent of a pollutant (in relative percent) from the non-SF to SF period. This methodology is fine for primary pollutants (such as NOx and BC), and it may also work for some secondary pollutants (perhaps $SO_4^=$); but it might not entirely applicable to $O_3$ since ozone production changes non-linearly with an increase (or a decrease) of NOx. A stronger evidence is needed before a conclusion on the origin of ozone is given in the manuscript.

Specific comments:

Line 12 and Line 49-53: During the SF, the power plants are generally not shutdown in the megacities of China. Are there any power plants in Shenzhen? Where the industrial area of Shenzhen located in reference to the measurement site? Are the wind roses of various pollutants in the SF different from those in the non-SF period?

Line 19-21: Here it might not be suitable to say "decreasing of regional pollutants" since there is an increase of $O_3$ by 6%.

Line 47: It might not be fully suitable to say so. The reported emission reductions could be verified by comparisons of different approaches, e.g., ground measurements, satellite observations, and model simulations with different emission inventories.

Line 77-79: Figure S1 provides only geographical position of measurement site. How far away from the main traffic roads the site is? How about regional distributions of pollutants' emissions?

Line 135: Figures S3-S5 provide more detailed information and can be moved into the formal part of the manuscript.

L174: CO is NOT a typical SECONDARY regional air pollutant as most of CO in the urban and polluted areas are mostly probably to be primarily emitted.

L183-184, L252-253, L261-267, L309-312 and L337-340: According to the study, isoprene and DMS measured in Shenzhen have anthropogenic sources. But they are repeatly described as "natural" gases; on the other hand, sometimes they are classified as "pollutants". These vague expressions should be corrected.

Line 191-192: It should noted that there is an increase of O3 to a small extent.

L195-202: Are there any power plants in Shenzhen and nearby areas? A plot of the regional emission distributions of SO2 would be helpful for the reader to follow the discussion here. It would also be great if the wind rose or trajectory analysis result could be given.

L212-215 and L300-301: The sentence "meteorology has only a small impact on their concentrations" is misleading. Note that only for the period average values are the result consistent.

L231, Figure 2: Ticks for 10 nm and 1000 nm could be given.

L288-289: In-depth analysis should be performed if you insist that the lack of NOx at noon during the SF period hinders the generation of O3. Figure 3B shows that the levels of NOx in the SF and NSFM at noon are comparable. It seems that the difference in VOCs might play an important role.

L300: What are the wind fields look like? It might be more appropriate to say the wind

field patterns are the same.

L368-370: The concept of regional air pollutants is unclear. How are they defined? Even for NOx, it can also result in regional pollution.

---

## Author Comment (AC1) · 6 Jun 2017

Referee #2's comments General comments: Large reduction in emissions of anthropogenic pollutants in a mega city during the week-long holiday gives an opportunity to investigate how the urban air quality reacts to reductions in emission. This study conducted a systematical examination of the "Spring Festival effect" over three consecutive winters in Shenzhen with a population of greater than ten million, and the difference in the concentrations of various air pollutants between the Spring Festival (SF) and non-Spring Festival (NSF) periods was exploited to indicate that the origins of pollutants are primarily local or regional.

[Figure]

Although "holiday effect" has been discussed in a large number of studies in recent decades, the rich and comprehensive dataset provided in this study is informative and helpful for understanding of sources of a large array of species with considerable differences in properties. The authors clearly show their own contribution in the study of Spring Festival effect. In the present form, the authors focus on exploiting the percent change in the concentrations of various air pollutants to differentiate contribution from local and regional sources. The value of this paper could be further enhanced if the authors can make more in-depth discussion on the species (e.g., PM0.8–2.5 and O3), which had a small difference between the SF and NSF periods. It is interesting to know that these species revealed only a small difference when traffic flow dropped by âĹij50% and the industrial plants were almost entirely shut down. It is well known that PM has numerous and complex constituents contributed from diversified sources (primary and secondary, anthropogenic and natural), and O3 is a secondary photochemical product with nonlinear relation with its precursors NOx and VOCs (anthropogenic and natural). I encourage the author to strengthen the link between their observation results and the possible causes (composition, property, sources, transport, physical and chemical processing, meteorology, etc.). In general, the manuscript was well written and organized. The subject of the paper is well within the scope of ACP. The paper is suitable for publication provided the general comments and following points are addressed.

Specific comments: 1. Page 7 Lines 137-139: 2016. Are "all fragments" of m/z 44 and m/z 57 measured by AMS are the tracers of oxygenated organic aerosol and primary hydrocarbon organic aerosol, respectively? Or "most of fragments" are?

REPLY: We have corrected in the revised manuscript as below: "Most of fragments of m/z 44 and m/z 57 are the tracer of oxygenated organic aerosol and the tracer of primary hydrocarbon organic aerosol (Zhang et al., 2005), respectively, which are measured by AMS."

2. Page 11 Lines 212-213: This sentence is confusing to me. In addition to emissions

and sinks, the concentration of air pollutants is also dominated by meteorological conditions, especially PBL and wind field. In Table 1, meteorological conditions in NSFT and NSFM are not much different (similar wind speed and no precipitation), and it may be the major reason for the small difference in the concentrations of most air pollutants between the two periods. It is not suitable to conclude that meteorology has only a small impact on their concentrations. Please provide a more appropriate interpretation.

REPLY: To be more rigorous, this sentence is changed to: "The decreasing ratios of various species during SF when compared with the NSFT and NSFM periods are similar, which suggests that the meteorological variations might not be the dominant reason for the species decreasing during SF."

3. Page 15 Lines 282-283: "emissions are greater on holidays than on non-holidays" and"In addition, emissions were higher during the 2008 Beijing Olympic Games" Do the authors mean "emissions of O3"? If so, I suggest using the word "concentrations" instead of "emissions".

REPLY: We have corrected in the revised manuscript.

4. Page 15 Lines 287 and 288: VOC/NOx ratio gives an important idea that it tends towards a NOx-sensitive or VOC-sensitive environment. However, VOCs are a very complex mixture of compounds with large difference in reactivity with respect to ozone formation. The true impact of VOCs to ozone formation is more relevant to the total reactivities of VOC species rather than to the total amount of VOCs. Furthermore, the VOCs reported in the study (Table S1, measured by PTR–MS) include only a total of 13 masses. TVOC/NOx ratio in the study should be carefully used to explain its effect on the ozone level due to above-mentioned concerns.

REPLY: We agree that based on limited VOCs measurements, it is not reasonable to get the conclusions that whether the O3 formation is NOx-sensitive or VOCs-sensitive. Therefore, in the revised manuscript, we only state as below: "As a result, although the

[Figure]

reduction in emissions of urban anthropogenic sources leads to a significant decline of NOx and VOCs, this reduction does not mitigate the average ambient O3 concentration, which implies that the concentration ratio VOC/NOx plays an important role in controlling O3 concentration."

5. Page 15 Lines 288 and 289: Do the authors mean the chemical regime at noon during the SF period was NOx-sensitive? If so, the authors should provide a clearer explanation to support the statement.

REPLY: Following the reply to the above question, we are not making the conclusion about whether the O3 formation is NOx-sensitive or VOCs-sensitive.

---

## Author Comment (AC2) · 6 Jun 2017

General comments: The paper presents an assessment of the effect of the Chinese Spring Festival on urban air quality in a southern China city. Various trace gases and aerosols were measured over three consecutive winters (2014-2016), including both the Spring Festival (SF) and non-Spring Festival (non-SF) periods, at an urban site of Shenzhen city. By comparing the concentrations of these pollutants in different periods, the authors show that the decreases for some pollutants are significant (by 50%-80%) in the SF with respect to the non-SF while the decreases of others pollutants are smaller. They suggest that such differences in the reduction extent of pollutant

concentrations be used to judge their sources, i.e., predominantly from local emissions or from regional transport. This study, together with a valuable dataset, should be a welcome addition to the literatures on the holiday effects on air quality. The manuscript can be accepted for publication after the following issues have been well addressed.

As the main purpose of the study is to distinguish the sources of urban pollution between the local emissions and regional transport, the local and regional pollution sources should have been clearly defined. It would be great if the authors could provide a map showing the geographical distributions of emissions rates of major air pollutants in Shenzhen city and its surrounding areas during wintertime. It is also suggested that if possible, the geographical extent of pollution reduction during the SF in relative to the non-SF period be identified, perhaps by using population density or satellite product.

The study identifies the primary source of each pollutant, i.e. the local or regional origin, using the reduction extent of a pollutant (in relative percent) from the non-SF to SF period. This methodology is fine for primary pollutants (such as NOx and BC), and it may also work for some secondary pollutants (perhaps $SO_4^{2-}$); but it might not entirely applicable to O3 since ozone production changes non-linearly with an increase (or a decrease) of NOx. A stronger evidence is needed before a conclusion on the origin of ozone is given in the manuscript.

Specific comments:

1. Line 12 and Line 49-53: During the SF, the power plants are generally not shutdown in the megacities of China. Are there any power plants in Shenzhen? Where the industrial area of Shenzhen located in reference to the measurement site? Are the wind roses of various pollutants in the SF different from those in the non-SF period?

REPLY: There are few power plants in Shenzhen. We now cite a paper (Zheng et al., 2009b) in section 2.1 to describe the relative location of the sampling site in terms of pollutant emissions in PRD, as below: "A highly resolved temporal and spatial emission inventory for PRD indicates that the sampling area is characterized by lower SO2

emissions but higher NOx and VOCs emissions in comparison with other areas in PRD (Zheng et al., 2009b)." Since the data points on each wind direction are limited, wind rose analysis may not be a solid evidence. However, the analysis of relationship between pollutant concentrations and wind speeds during SF and NSF in section 3.3 can already well reveal that Group LD was highly influenced by wind speed.

2. Line 19-21: Here it might not be suitable to say "decreasing of regional pollutants" since there is an increase of O3 by 6%.

REPLY: We rephrased the sentences by"The concentration variation of species mostly from regional or natural sources, however, is found to be much less, such as for bulk PM2.5."

3. Line 47: It might not be fully suitable to say so. The reported emission reductions could be verified by comparisons of different approaches, e.g., ground measurements, satellite observations, and model simulations with different emission inventories.

REPLY: ". . .so the reported emission reductions cannot be verified" changed to ". . .so the air quality monitoring campaigns cannot be repeated."

4. Line 77-79: Figure S1 provides only geographical position of measurement site. How far away from the main traffic roads the site is? How about regional distributions of pollutants' emissions?

REPLY: We have provided the relevant information in the revised text as below: "PKUSZ is located in the western urban area of Shenzhen, and there are no significant anthropogenic pollution sources nearby except a local road ∼100 m far from the sampling site. A highly resolved temporal and spatial emission inventory for PRD indicates that the sampling area is characterized by lower SO2 emissions but higher NOx and VOCs emissions in comparison with other areas in PRD (Zheng et al., 2009b)."

5. Line 135: Figures S3-S5 provide more detailed information and can be moved into the formal part of the manuscript.

REPLY: After careful consideration, we still believe that Figures S3-S5 are too large to be moved into the main text. Therefore, supporting information could be a better choice when considering that ACP is freely available on the internet.

6. L174: CO is NOT a typical SECONDARY regional air pollutant as most of CO in the urban and polluted areas are mostly probably to be primarily emitted.

REPLY: It is a typo. "secondary" is now deleted.

7. L183-184, L252-253, L261-267, L309-312 and L337-340: According to the study, isoprene and DMS measured in Shenzhen have anthropogenic sources. But they are repeatly described as "natural" gases; on the other hand, sometimes they are classified as "pollutants". These vague expressions should be corrected.

REPLY: We have checked all the manuscript and rephrased the words to say that they are "mainly emitted by natural sources"

8. Line 191-192: It should noted that there is an increase of O3 to a small extent.

REPLY: We have rephrased the sentences as below: "The group of pollutants with smallest decrease in concentration (hereinafter called "SD") includes SO2, PM0.8–2.5, and O3 (8h) in the case of comparison with NSFM. The magnitude of the average percent change is less than 20% relative to the two NSF periods. It is interesting to note that there was even concentration increase in other O3–related cases."

9. L195-202: Are there any power plants in Shenzhen and nearby areas? A plot of the regional emission distributions of SO2 would be helpful for the reader to follow the discussion here. It would also be great if the wind rose or trajectory analysis result could be given.

REPLY: There are few power plants in Shenzhen. We now cite a paper (Zheng et al., 2009b) in section 2.1 to describe the relative location of the sampling site in terms of pollutant emissions in PRD, as below: "A highly resolved temporal and spatial emission inventory for PRD indicates that the sampling area is characterized by lower SO2

emissions but higher NOx and VOCs emissions in comparison with other areas in PRD (Zheng et al., 2009b)." Since the data points on each wind direction are limited, wind rose analysis may not be a solid evidence. However, the analysis of relationship between pollutant concentrations and wind speeds during SF and NSF in section 3.3 can already well reveal that Group LD was highly influenced by wind speed.

10. L212-215 and L300-301: The sentence "meteorology has only a small impact on their concentrations" is misleading. Note that only for the period average values are the result consistent.

REPLY: To be more rigorous, this sentence is changed to: "The decreasing ratios of various species during SF when compared with the NSFT and NSFM periods are similar, which suggests that the meteorological variations might not be the dominant reason for the species decreasing during SF."

11. L231, Figure 2: Ticks for 10 nm and 1000 nm could be given.

REPLY: Our instrument (SMPS) determined the particle number size distribution in the size range of 15–615 nm.

12. L288-289: In-depth analysis should be performed if you insist that the lack of NOx at noon during the SF period hinders the generation of O3. Figure 3B shows that the levels of NOx in the SF and NSFM at noon are comparable. It seems that the difference in VOCs might play an important role.

REPLY: We agree that based on limited VOCs measurements, it is not reasonable to get the conclusions that whether the O3 formation is NOx-sensitive or VOCs-sensitive. Therefore, in the revised manuscript, we only state as below: "As a result, although the reduction in emissions of urban anthropogenic sources leads to a significant decline of NOx and VOCs, this reduction does not mitigate the average ambient O3 concentration, which implies that the concentration ratio VOC/NOx plays an important role in controlling O3 concentration."

13. L300: What are the wind fields look like? It might be more appropriate to say the wind field patterns are the same.

REPLY: The wind rose plots can be seen in Figure S2. We take this suggestion to use "wind field patterns"

14. L368-370: The concept of regional air pollutants is unclear. How are they defined? Even for NOx, it can also result in regional pollution.

REPLY: In this study, we studied urban air quality and urban emissions, and thus regional pollutants refer to species from outside the urban areas. Yes, NOx can also be from regional transport. However, based on our results, NOx is found to be mostly from local urban emissions. In the revised text, we have added the words like "air pollutants mostly from regional or natural sources" to be more rigorous.

---

## Editor Comment (EC1) · J. Ma (Editor) · 7 Jun 2017

Dear Dr. Huang, Please consider the general comments from the referees carefully and give your specific responses when you submit your revised manuscript. Sincerely, Jianzhong Ma

---

## Author Comment (AC3) · 7 Jun 2017

Large reduction in emissions of anthropogenic pollutants in a mega city during the week-long holiday gives an opportunity to investigate how the urban air quality reacts to reductions in emission. This study conducted a systematical examination of the "Spring Festival effect" over three consecutive winters in Shenzhen with a population of greater than ten million, and the difference in the concentrations of various air pollutants between the Spring Festival (SF) and non-Spring Festival (NSF) periods was exploited to indicate that the origins of pollutants are primarily local or regional.

Although "holiday effect" has been discussed in a large number of studies in recent

decades, the rich and comprehensive dataset provided in this study is informative and helpful for understanding of sources of a large array of species with considerable differences in properties. The authors clearly show their own contribution in the study of Spring Festival effect. In the present form, the authors focus on exploiting the percent change in the concentrations of various air pollutants to differentiate contribution from local and regional sources. The value of this paper could be further enhanced if the authors can make more in-depth discussion on the species (e.g., PM0.8–2.5 and O3), which had a small difference between the SF and NSF periods. It is interesting to know that these species revealed only a small difference when traffic flow dropped by âLij50% and the industrial plants were almost entirely shut down. It is well known that PM has numerous and complex constituents contributed from diversified sources (primary and secondary, anthropogenic and natural), and O3 is a secondary photochemical product with nonlinear relation with its precursors NOx and VOCs (anthropogenic and natural). I encourage the author to strengthen the link between their observation results and the possible causes (composition, property, sources, transport, physical and chemical processing, meteorology, etc.). In general, the manuscript was well written and organized. The subject of the paper is well within the scope of ACP. The paper is suitable for publication provided the general comments and following points are addressed.

REPLY: We have looked for new useful evidence to explain the small decrease of SO2 and PM0.8–2.5 in Group SD, and have given new information as below in section 3.1:

"On the other hand, a piece of evidence for the regional origin of SO2 is from the newly established 356 m meteorological and environmental monitoring iron tower in Shenzhen. The ambient SO2 concentrations were similar at the highest platform (ave.=7.4 ppbv@325 m) and the lowest platform (ave.=7.2 ppbv@60 m) during January–February, 2017, indicating that SO2 was already well mixed in the atmosphere and the local contributions should be minor. In contrast, the concentrations of NOx, which belongs to Group LD, had a 56% higher concentration at the lowest platform

than at the highest platform (Zhuang, 2017). The small decrease of SO2 is thus a reasonable result of the stable emissions during the SF periods and the primarily regional origin."

"In terms of chemical composition of PM0.8–2.5, implications can be found in our previous size distribution measurement of aerosol chemical composition, using a ten-stage micro orifice uniform deposit impactor (MOUDI), during the fall to winter in Shenzhen (Lan et al., 2011). The results clearly indicate that smaller fine particles (e.g., 0.18–0.56 $\mu$m) contains relatively more BC (BC/SO42-=0.83), while larger fine particles (e.g., 1.0–1.8 $\mu$m) contained a higher proportion of SO42- (BC/SO42-=0.17). The SO42- in PM2.5 in Shenzhen has been well proved to be mostly a regional pollutant, with similar concentrations at various sites including both urban and rural sites (Huang et al., 2014). Therefore, the very small decrease of PM0.8–2.5 during SF should be closely related to its enrichment of secondary regional species like SO42-."

As to the unique variation of O3, it is clearly related to the nonlinear relation with its precursors NOx and VOCs, as stated by this reviewer. The other reviewer also pointed out that, based on limited VOCs measured, we cannot achieve a conclusion that whether the O3 formation is VOCs-sensitive or NOx sensitive. Therefore, in the revised manuscript, we only pointed out the idea as below in section 3.2:

". . .As a result, although the reduction in emissions of urban anthropogenic sources leads to a large decline of NOx and VOCs, this reduction does not mitigate the average ambient O3 concentration, which implies that the concentration ratio VOCs/NOx play an important role in controlling O3 concentration."

---

## Author Comment (AC4) · 7 Jun 2017

General comments: The paper presents an assessment of the effect of the Chinese Spring Festival on urban air quality in a southern China city. Various trace gases and aerosols were measured over three consecutive winters (2014-2016), including both the Spring Festival (SF) and non-Spring Festival (non-SF) periods, at an urban site of Shenzhen city. By comparing the concentrations of these pollutants in different periods, the authors show that the decreases for some pollutants are significant (by 50%-80%) in the SF with respect to the non-SF while the decreases of others pollutants are smaller. They suggest that such differences in the reduction extent of pollutant

concentrations be used to judge their sources, i.e., predominantly from local emissions or from regional transport. This study, together with a valuable dataset, should be a welcome addition to the literatures on the holiday effects on air quality. The manuscript can be accepted for publication after the following issues have been well addressed.

As the main purpose of the study is to distinguish the sources of urban pollution between the local emissions and regional transport, the local and regional pollution sources should have been clearly defined. It would be great if the authors could provide a map showing the geographical distributions of emissions rates of major air pollutants in Shenzhen city and its surrounding areas during wintertime. It is also suggested that if possible, the geographical extent of pollution reduction during the SF in relative to the non-SF period be identified, perhaps by using population density or satellite product.

The study identifies the primary source of each pollutant, i.e. the local or regional origin, using the reduction extent of a pollutant (in relative percent) from the non-SF to SF period. This methodology is fine for primary pollutants (such as NOx and BC), and it may also work for some secondary pollutants (perhaps $SO_4^{2-}$); but it might not entirely applicable to O3 since ozone production changes non-linearly with an increase (or a decrease) of NOx. A stronger evidence is needed before a conclusion on the origin of ozone is given in the manuscript.

REPLY: As to the definition of the local and regional pollution sources, please see our reply to your specific comment #14. In the revised manuscript, we have tried to define them as below: "Apparently, the dominant sources for most of these pollutants are primarily local emissions in the urban scale, such as combustion sources for BC, m/z 57, and NOx. . ."; "The species in this group are either typical regional air pollutants mostly from beyond the urban scale, such as CO, which has a long lifetime. . ."

As to a map showing the geographical distributions of emissions rates of major air pollutants and the geographical extent of pollution reduction during the SF, they need a lot of supporting data sources, which are not easily available, and such work could be

much beyond the scope of this paper. Alternatively, we have cited a paper of emission inventory in PRD (Zheng et al., 2009b) in section 2.1 to describe the relative location of the sampling site in terms of pollutant emissions in PRD, as below: "A highly resolved temporal and spatial emission inventory for PRD indicates that the sampling area is characterized by lower SO2 emissions but higher NOx and VOCs emissions in comparison with other areas in PRD (Zheng et al., 2009b)."

As to the conclusion on the origin of ozone, we agree on your specific comment #12 that based on limited VOCs measurements, it is not reasonable to get the conclusions that whether the O3 formation is NOx-sensitive or VOCs-sensitive. Therefore, in the revised manuscript, we only state as below: "As a result, although the reduction in emissions of urban anthropogenic sources leads to a significant decline of NOx and VOCs, this reduction does not mitigate the average ambient O3 concentration, which implies that the concentration ratio VOC/NOx plays an important role in controlling O3 concentration."

---

## Author Response (AR3)

**Co-Editor's comments**

Here I have a question on the use of a phrase in the Abstract (Page 1, line 20). What do you mean by "natural pollutants"? Please give a specification or correction.

**5 REPLY:**

Corrected to "species emitted from natural sources".

**Referee #1's comments**

\_\_\_\_\_

**10**

1. It is difficult to get the point from the title of this paper. What does the "Characterization of local and regional sources" mean? It is not clear though one finish reading the paper.

**REPLY:**

The title has been changed to "Differentiating local and regional sources of Chinese urban air pollution based on effect of Spring Festival", which may make the idea of this paper clearer.

2. Page 1 Line 22-23: "More detailed analysis----- reduce urban  $PM_{2.5}$  levels in China". Are you sure that the results from this paper can be used to all China? The air pollution problems are different in various regions in China. So the results from Shenzhen cannot represent the normal situation in other regions.

**20 REPLY:**

To be more rigorous, this sentence is changed to "More detailed analysis of the Spring Festival effect reveals an urgent need to reduce emissions of  $SO_2$  and VOCs on a regional scale rather than on an urban scale to reduce urban  $PM_{2.5}$  in Shenzhen, which can also produce some use for reference for other megacities in China."

25 3. Introduction: The previous studies are not enough. More references should be added.

**REPLY:**

We have rephrased the introduction part, with more details and references added as below (see the red words):

[revised manuscript text omitted]

4. Page 3 Line 47 and Line 52: Shenzhen is a megacity in South China. In Spring Festival, over 50% of the residents are used to travel back to their hometown. But in other small or medium city, the situation are total opposite. The representativeness should be considered.

**REPLY:**

5 Yes, the SF effect is a feature for Shenzhen rather than other cities, which is the reason why we selected Shenzhen as the area for this unique study. We have rephrased the few sentences to make it clearer that the SF study is special for Shenzhen.

5. Page 15 Line 289-290: It is very important that the concentration ratio VOC/NOx plays a role in controlling  $O_3$  concentration. This result is very useful to make sure the control strategy of  $O_3$ . More discussions should be added.

**10 **REPLY:**

Following the editor's requirement, this comment will be addressed after the open discussion of the current version of the paper.

6. Page 16 Fig.3V: It is interesting that the super fine particle did not decrease due to the Spring Festival. The emission sources

15 and formation process of these super fine particle are important to the controlling the air pollution problem. More discussion should be emphasized.

**REPLY:**

Following the editor's requirement, this comment will be addressed after the open discussion of the current version of the paper.

**20**

7. Page 17 Line 319-320: It seems that the SF period, the regional photochemical production is not weakened. The reason should be discussed more.

**REPLY:**

Following the editor's requirement, this comment will be addressed after the open discussion of the current version of the paper.
* * *
**Referee #2's comments** No detailed comments.

30

35

**Referee #2's comments**

General comments:

Large reduction in emissions of anthropogenic pollutants in a mega city during the week-long holiday gives an opportunity to investigate how the urban air quality reacts to reductions in emission. This study conducted a systematical examination of the

- 5 "Spring Festival effect" over three consecutive winters in Shenzhen with a population of greater than ten million, and the difference in the concentrations of various air pollutants between the Spring Festival (SF) and non-Spring Festival (NSF) periods was exploited to indicate that the origins of pollutants are primarily local or regional.
- Although "holiday effect" has been discussed in a large number of studies in recent decades, the rich and comprehensive dataset provided in this study is informative and helpful for understanding of sources of a large array of species with considerable differences in properties. The authors clearly show their own contribution in the study of Spring Festival effect. In the present form, the authors focus on exploiting the percent change in the concentrations of various air pollutants to differentiate contribution from local and regional sources. The value of this paper could be further enhanced if the authors can make more in-depth discussion on the species (e.g., PM0.8–2.5 and O3), which had a small difference between the SF and NSF
- 15 periods. It is interesting to know that these species revealed only a small difference when traffic flow dropped by  $\sim$ 50% and the industrial plants were almost entirely shut down. It is well known that PM has numerous and complex constituents contributed from diversified sources (primary and secondary, anthropogenic and natural), and O3 is a secondary photochemical product with nonlinear relation with its precursors NOx and VOCs (anthropogenic and natural). I encourage the author to strengthen the link between their observation results and the possible causes (composition, property, sources, transport,
- 20 physical and chemical processing, meteorology, etc.). In general, the manuscript was well written and organized. The subject of the paper is well within the scope of ACP. The paper is suitable for publication provided the general comments and following points are addressed.

**REPLY:**

25 We have looked for new useful evidence to explain the small decrease of  $SO_2$  and  $PM_{0.8-2.5}$  in Group SD, and have given new information as below in section 3.1:

"On the other hand, a piece of evidence for the regional origin of  $SO_2$  is from the newly established 356 m meteorological and environmental monitoring iron tower in Shenzhen. The ambient  $SO_2$  concentrations were similar at the highest platform

- 30 (ave.=7.4 ppbv@325 m) and the lowest platform (ave.=7.2 ppbv@60 m) during January–February, 2017, indicating that SO2 was already well mixed in the atmosphere and the local contributions should be minor. In contrast, the concentrations of NOx, which belongs to Group LD, had a 56% higher concentration at the lowest platform than at the highest platform (Zhuang, 2017). The small decrease of SO2 is thus a reasonable result of the stable emissions during the SF periods and the primarily regional origin."
- 35

40

"In terms of chemical composition of  $PM_{0.8-2.5}$ , implications can be found in our previous size distribution measurement of aerosol chemical composition, using a ten-stage micro orifice uniform deposit impactor (MOUDI), during the fall to winter in Shenzhen (Lan et al., 2011). The results clearly indicate that smaller fine particles (e.g., 0.18–0.56 µm) contains relatively more BC (BC/SO42–=0.83), while larger fine particles (e.g., 1.0–1.8 µm) contained a higher proportion of SO42– (BC/SO42–=0.17). The SO42– in PM2.5 in Shenzhen has been well proved to be mostly a regional pollutant, with similar concentrations at various sites including both urban and rural sites (Huang et al., 2014). Therefore, the very small decrease of PM0.8–2.5 during

SF should be closely related to its enrichment of secondary regional species like SO42-."

As to the unique variation of O3, it is clearly related to the nonlinear relation with its precursors NOx and VOCs, as stated by

45 this reviewer. The other reviewer also pointed out that, based on limited VOCs measured, we cannot achieve a conclusion that whether the  $O_3$  formation is VOCs-sensitive or NOx sensitive. Therefore, in the revised manuscript, we only pointed out the idea as below in section 3.2:

"... As a result, although the reduction in emissions of urban anthropogenic sources leads to a large decline of NOx and VOCs, this reduction does not mitigate the average ambient  $O_3$  concentration, which implies that the concentration ratio VOCs/NOx play an important role in controlling O3 concentration."

5 Specific comments:

1. Page 7 Lines 137-139: 2016. Are "all fragments" of m/z 44 and m/z 57 measured by AMS are the tracers of oxygenated organic aerosol and primary hydrocarbon organic aerosol, respectively? Or "most of fragments" are?

**REPLY:**

We have corrected in the revised manuscript as below:

"Most of fragments of m/z 44 and m/z 57 are the tracer of oxygenated organic aerosol and the tracer of primary hydrocarbon 10 organic aerosol (Zhang et al., 2005), respectively, which are measured by AMS."

2. Page 11 Lines 212-213: This sentence is confusing to me. In addition to emissions and sinks, the concentration of air pollutants is also dominated by meteorological conditions, especially PBL and wind field. In Table 1, meteorological conditions

15 in NSFT and NSFM are not much different (similar wind speed and no precipitation), and it may be the major reason for the small difference in the concentrations of most air pollutants between the two periods. It is not suitable to conclude that meteorology has only a small impact on their concentrations. Please provide a more appropriate interpretation.

**REPLY:**

To be more rigorous, this sentence is changed to:

"The decreasing ratios of various species during SF when compared with the NSFT and NSFM periods are similar, which 20 suggests that the meteorological variations might not be the dominant reason for the species decreasing during SF."

3. Page 15 Lines 282-283: "emissions are greater on holidays than on non-holidays" and "In addition, emissions were higher during the 2008 Beijing Olympic Games" Do the authors mean "emissions of O3"? If so, I suggest using the word "concentrations" instead of "emissions".

**REPLY:**

25

We have corrected in the revised manuscript.

4. Page 15 Lines 287 and 288: VOC/NOx ratio gives an important idea that it tends towards a NOx-sensitive or VOC-sensitive 30 environment. However, VOCs are a very complex mixture of compounds with large difference in reactivity with respect to ozone formation. The true impact of VOCs to ozone formation is more relevant to the total reactivities of VOC species rather than to the total amount of VOCs. Furthermore, the VOCs reported in the study (Table S1, measured by PTR-MS) include only a total of 13 masses. TVOC/NOx ratio in the study should be carefully used to explain its effect on the ozone level due to above-mentioned concerns.

**35 **REPLY:**

We agree that based on limited VOCs measurements, it is not reasonable to get the conclusions that whether the  $O_3$  formation is NOx-sensitive or VOCs-sensitive. Therefore, in the revised manuscript, we only state as below:

"As a result, although the reduction in emissions of urban anthropogenic sources leads to a significant decline of NOx and VOCs, this reduction does not mitigate the average ambient  $O_3$  concentration, which implies that the concentration ratio VOC/NOx plays an important role in controlling O3 concentration."

5. Page 15 Lines 288 and 289: Do the authors mean the chemical regime at noon during the SF period was NOx-sensitive? If so, the authors should provide a clearer explanation to support the statement.

**REPLY:**

45 Following the reply to the above question, we are not making the conclusion about whether the O3 formation is NOx-sensitive or VOCs-sensitive.

**-----**

**Referee #3's comments**

General comments:

- The paper presents an assessment of the effect of the Chinese Spring Festival on urban air quality in a southern China city.
  Various trace gases and aerosols were measured over three consecutive winters (2014-2016), including both the Spring Festival (SF) and non-Spring Festival (non-SF) periods, at an urban site of Shenzhen city. By comparing the concentrations of these pollutants in different periods, the authors show that the decreases for some pollutants are significant (by 50%-80%) in the SF with respect to the non-SF while the decreases of others pollutants are smaller. They suggest that such differences in the reduction extent of pollutant concentrations be used to judge their sources, i.e., predominantly from local emissions or from
- 10 regional transport. This study, together with a valuable dataset, should be a welcome addition to the literatures on the holiday effects on air quality. The manuscript can be accepted for publication after the following issues have been well addressed.

As the main purpose of the study is to distinguish the sources of urban pollution between the local emissions and regional transport, the local and regional pollution sources should have been clearly defined. It would be great if the authors could

15 provide a map showing the geographical distributions of emissions rates of major air pollutants in Shenzhen city and its surrounding areas during wintertime. It is also suggested that if possible, the geographical extent of pollution reduction during the SF in relative to the non-SF period be identified, perhaps by using population density or satellite product.

The study identifies the primary source of each pollutant, i.e. the local or regional origin, using the reduction extent of a

20 pollutant (in relative percent) from the non-SF to SF period. This methodology is fine for primary pollutants (such as NOx and BC), and it may also work for some secondary pollutants (perhaps  $SO_4^{2-}$ ); but it might not entirely applicable to  $O_3$  since ozone production changes non-linearly with an increase (or a decrease) of NOx. A stronger evidence is needed before a conclusion on the origin of ozone is given in the manuscript.

**REPLY:**

25 As to the definition of the local and regional pollution sources, please see our reply to your specific comment #14. In the revised manuscript, we have tried to define them as below:

"Apparently, the dominant sources for most of these pollutants are primarily local emissions in the urban scale, such as combustion sources for BC, m/z 57, and NOx..."; "The species in this group are either typical regional air pollutants mostly from beyond the urban scale, such as CO, which has a long lifetime..."

**30**

As to a map showing the geographical distributions of emissions rates of major air pollutants and the geographical extent of pollution reduction during the SF, they need a lot of supporting data sources, which are not easily available, and such work could be much beyond the scope of this paper. Alternatively, we have cited a paper of emission inventory in PRD (Zheng et al., 2009b) in section 2.1 to describe the relative location of the sampling site in terms of pollutant emissions in PRD, as below:

- 35 "A highly resolved temporal and spatial emission inventory for PRD indicates that the sampling area is characterized by lower SO2 emissions but higher NOx and VOCs emissions in comparison with other areas in PRD (Zheng et al., 2009b)." As to the conclusion on the origin of ozone, we agree on your specific comment #12 that based on limited VOCs measurements, it is not reasonable to get the conclusions that whether the O3 formation is NOx-sensitive or VOCs-sensitive. Therefore, in the revised manuscript, we only state as below:
- 40 "As a result, although the reduction in emissions of urban anthropogenic sources leads to a significant decline of NOx and VOCs, this reduction does not mitigate the average ambient O3 concentration, which implies that the concentration ratio VOC/NOx plays an important role in controlling O3 concentration."

Specific comments:

45 1. Line 12 and Line 49-53: During the SF, the power plants are generally not shutdown in the megacities of China. Are there any power plants in Shenzhen? Where the industrial area of Shenzhen located in reference to the measurement site? Are the wind roses of various pollutants in the SF different from those in the non-SF period?

**REPLY:**

There are few power plants in Shenzhen. We now cite a paper (Zheng et al., 2009b) in section 2.1 to describe the relative location of the sampling site in terms of pollutant emissions in PRD, as below:

"A highly resolved temporal and spatial emission inventory for PRD indicates that the sampling area is characterized by lower SO2 emissions but higher NOx and VOCs emissions in comparison with other areas in PRD (Zheng et al., 2009b)." Since the data points on each wind direction are limited, wind rose analysis may not be a solid evidence. However, the analysis

of relationship between pollutant concentrations and wind speeds during SF and NSF in section 3.3 can already well reveal that Group LD was highly influenced by wind speed.

5

2. Line 19-21: Here it might not be suitable to say "decreasing of regional pollutants" since there is an increase of  $O_3$  by 6%. **REPLY:**

We rephrased the sentences by "The concentration variation of species mostly from regional or natural sources, however, is found to be much less, such as for bulk PM25." 10

3. Line 47: It might not be fully suitable to say so. The reported emission reductions could be verified by comparisons of different approaches, e.g., ground measurements, satellite observations, and model simulations with different emission inventories.

**15 **REPLY:**

"...so the reported emission reductions cannot be verified" changed to "...so the air quality monitoring campaigns cannot be repeated."

4. Line 77-79: Figure S1 provides only geographical position of measurement site. How far away from the main traffic roads the site is? How about regional distributions of pollutants' emissions?

**REPLY:**

20

We have provided the relevant information in the revised text as below:

"PKUSZ is located in the western urban area of Shenzhen, and there are no significant anthropogenic pollution sources nearby except a local road ~100 m far from the sampling site. A highly resolved temporal and spatial emission inventory for PRD indicates that the sampling area is characterized by lower SO2 emissions but higher NOx and VOCs emissions in comparison

25 with other areas in PRD (Zheng et al., 2009b)."

5. Line 135: Figures S3-S5 provide more detailed information and can be moved into the formal part of the manuscript. **REPLY:**

30 After careful consideration, we still believe that Figures S3-S5 are too large to be moved into the main text. Therefore, supporting information could be a better choice when considering that ACP is freely available on the internet.

6. L174: CO is NOT a typical SECONDARY regional air pollutant as most of CO in the urban and polluted areas are mostly probably to be primarily emitted.

**35 **REPLY:**

It is a typo. "secondary" is now deleted.

7. L183-184, L252-253, L261-267, L309-312 and L337-340: According to the study, isoprene and DMS measured in Shenzhen have anthropogenic sources. But they are repeatly described as "natural" gases; on the other hand, sometimes they are classified

**40 as "pollutants". These vague expressions should be corrected.**

**REPLY:**

We have checked all the manuscript and rephrased the words to say that they are "mainly emitted by natural sources"

8. Line 191-192: It should noted that there is an increase of  $O_3$  to a small extent.

**45 **REPLY:**

We have rephrased the sentences as below:

"The group of pollutants with smallest decrease in concentration (hereinafter called "SD") includes SO2,  $PM_{0.8-2.5}$ , and  $O_3$  (8h) in the case of comparison with NSFM. The magnitude of the average percent change is less than 20% relative to the two NSF periods. It is interesting to note that there was even concentration increase in other O3-related cases."

9. L195-202: Are there any power plants in Shenzhen and nearby areas? A plot of the regional emission distributions of  $SO_2$  would be helpful for the reader to follow the discussion here. It would also be great if the wind rose or trajectory analysis result could be given.

**REPLY:**

5 There are few power plants in Shenzhen. We now cite a paper (Zheng et al., 2009b) in section 2.1 to describe the relative location of the sampling site in terms of pollutant emissions in PRD, as below:

"A highly resolved temporal and spatial emission inventory for PRD indicates that the sampling area is characterized by lower SO2 emissions but higher NOx and VOCs emissions in comparison with other areas in PRD (Zheng et al., 2009b)."

- Since the data points on each wind direction are limited, wind rose analysis may not be a solid evidence. However, the analysis
- 10 of relationship between pollutant concentrations and wind speeds during SF and NSF in section 3.3 can already well reveal that Group LD was highly influenced by wind speed.

10. L212-215 and L300-301: The sentence "meteorology has only a small impact on their concentrations" is misleading. Note that only for the period average values are the result consistent.

**15 REPLY:**

To be more rigorous, this sentence is changed to:

"The decreasing ratios of various species during SF when compared with the NSFT and NSFM periods are similar, which suggests that the meteorological variations might not be the dominant reason for the species decreasing during SF."

20 11. L231, Figure 2: Ticks for 10 nm and 1000 nm could be given.

**REPLY:**

Our instrument (SMPS) determined the particle number size distribution in the size range of 15-615 nm.

12. L288-289: In-depth analysis should be performed if you insist that the lack of NOx at noon during the SF period hinders

25 the generation of O3. Figure 3B shows that the levels of NOx in the SF and NSFM at noon are comparable. It seems that the difference in VOCs might play an important role.

**REPLY:**

We agree that based on limited VOCs measurements, it is not reasonable to get the conclusions that whether the  $O_3$  formation is NOx-sensitive or VOCs-sensitive. Therefore, in the revised manuscript, we only state as below:

30 "As a result, although the reduction in emissions of urban anthropogenic sources leads to a significant decline of NOx and VOCs, this reduction does not mitigate the average ambient  $O_3$  concentration, which implies that the concentration ratio VOC/NOx plays an important role in controlling  $O_3$  concentration."

13. L300: What are the wind fields look like? It might be more appropriate to say the wind field patterns are the same.

**35 REPLY:**

The wind rose plots can be seen in Figure S2. We take this suggestion to use "wind field patterns"

14. L368-370: The concept of regional air pollutants is unclear. How are they defined? Even for NOx, it can also result in regional pollution.

**40 **REPLY:**

In this study, we studied urban air quality and urban emissions, and thus regional pollutants refer to species from outside the urban areas. Yes, NOx can also be from regional transport. However, based on our results, NOx is found to be mostly from local urban emissions. In the revised text, we have added the words like "air pollutants mostly from regional or natural sources" to be more rigorous.

**Co-editor's comments:**

Please consider the suggestion from the referee carefully and make necessary revisions and/or specific response. Alternatively, you may also consider using satellite data, e.g., OMI NO2, OMI SO2 and nighttime lights, to show the regional distributions of pollution in the investigated city and its surrounding areas.

**5 REPLY:**

Please see the reply to the referee. We have used the map of nightlights.

**-----**

**Referee#2's comments**

- 10 The revised manuscript does improve to some degree with ambiguous expressions corrected. Whiles the main purpose of this study is to distinguish the sources of urban pollution between the local emissions and regional transport, measurements were made only at an urban site. Therefore, I think the geographical distributions of emissions can be shown as compensation. There are some global and regional emission inventories with high resolution (0.5 degree by 0.5 degree) available, for example, the GEIA-ACCENT emission data (http://accent.aero.jussieu.fr/index.php), the Regional Emission inventory in ASia (REAS)
- 15 Data (https://www.nies.go.jp/REAS/), and the MEIC dataset from the Chinese research group. It might be true that none of these emission inventories provide information on the emission reduction during the festival. But a seasonal or annual average distribution of "normal" emissions will also be helpful for understanding the observational results. REPLY:

We have created a new Figure S1 to clearly indicate the regional pollution distribution of PRD and the location of the sampling

20 site, as below. In Line 79, we have also added this sentence "Figure S1 shows that the sampling area had a middle-level aerosol pollution during the winter in comparison with other areas in PRD and further areas."

Figure S1 The locations of Shenzhen and its sampling site. The upper map is part of the global maps of Earth at night from NASA (https://www.nasa.gov/feature/goddard/2017/new-night-lights-maps-open-up-possible-real-time-applications), which

25 indicates that PRD is one of the most developed regions in China. The lower map is the surface extinction coefficient (SEC) of dry aerosol in southern China in winter (using the monthly average in February, 2015 as the example) from the Institute for the Kong University of Science and Technology the Environment, Hong (http://envf.ust.hk/itfsi/index.py?where=Home/SAMap&id=premap), which presents the general spatial distribution of aerosol pollution, as well as the locations of Shenzhen and its sampling site.